# Disseminated Histoplasmosis in an Immunocompetent Patient from Southern Arizona

**DOI:** 10.3390/jof10110756

**Published:** 2024-10-31

**Authors:** Monique Crawford, Troy Weinstein, Alexis Elliott, Robert Klein, Michael Lee, Conner Reynolds, Talha Riaz

**Affiliations:** 1The University of Arizona College of Medicine, Tucson, AZ 85724, USA; moniqueramirez@arizona.edu (M.C.); tweinstein@arizona.edu (T.W.); 2Department of Pathology, The University of Arizona College of Medicine, Tucson, AZ 85724, USA; aselliott@arizona.edu (A.E.);; 3Department of Medical Imaging, The University of Arizona College of Medicine, Tucson, AZ 85724, USA; leem1ke@radiology.arizona.edu (M.L.); conner.reynolds@arizona.edu (C.R.); 4Division of Infectious Diseases, Department of Medicine, The University of Arizona College of Medicine, Tucson, AZ 85724, USA

**Keywords:** disseminated histoplasmosis, disseminated histoplasmosis in immunocompetent, treatment of histoplasmosis, endemic fungi, itraconazole levels

## Abstract

In this review, we present a case report of an immunocompetent 58-year-old male who presented with disseminated histoplasmosis (DH) outside of the known endemic regions. Due to his atypical clinical presentation that did not fit the classical clinical picture of DH, the diagnosis was delayed. In addition, we researched DH in the immunocompetent hosts as these cases are not common and leave the patient population vulnerable to delayed diagnosis. The literature supports considering histoplasmosis in the differential diagnosis among patients who present with possible exposures outside of endemic regions and are immunocompetent.

## 1. Introduction

*Histoplasma capsulatum* is a soil-based dimorphic fungus endemic to the Ohio and Mississippi River valleys and the Southeastern region of the United States. It is a yeast at body temperature, and at ambient temperatures, it is a mold. It is acquired via inhalation of microconidia that become lodged in the alveoli and engulfed by neutrophils and macrophages. Once in the body, the warmer temperature prompts its transformation into unicellular yeast forms 2–5 μm in diameter. In the lungs, the yeast phase is responsible for its infectious properties. As a result, the body forms organized granulomas to control the infection [1]. Clinical manifestations can range from subclinical and mild respiratory illness in the immunocompetent or chronic pulmonary histoplasmosis in those with underlying lung disease. The immunocompromised can present with progressive life-threatening disseminated histoplasmosis (DH) due to weakened cell-mediated immunity and impaired cytokine production [2]. Because it is usually asymptomatic in immunocompetent patients, the concern for DH is placed on patients who are immunocompromised. Original population studies by the Public Health Service showed over 80% of young asymptomatic adults living in Ohio and Mississippi River valleys have had previous infections with *H. capsulatum* as determined by skin antigen testing [3]. The spectrum of disease is determined by the inhaled conidia load and host immune competence [4].

Recent research has suggested that more cases are being seen outside of endemic regions, increasing the concern for a missed or delayed diagnosis. Even with a thorough social and travel history to determine possible exposures, it can be challenging to consider histoplasmosis as a diagnosis if the patient does not fit the typical clinical picture and has not visited or resided in endemic regions. In this review, we present an immunocompetent patient from Southern Arizona who presented with DH likely acquired from exposure to bat guano in Southern Arizona.

## 2. Case Presentation

The patient is a 58-year-old male with a history of tobacco use who was transferred from an outside hospital due to concerns for malignancy involving a left adrenal mass. He reported 1.5 months of progressive, generalized weakness, fatigue, decreased appetite, and an unintentional weight loss of >100 lbs. He was found to have an adrenal mass as part of a work-up for abdominal pain at an outside hospital. The patient currently lives close to cattle auction farms in southern Arizona and reported significant bat guano on his property. Although the patient has not left Arizona in the past 30 years, other pertinent social history includes residence in the Pacific Northwest, employment as a smokejumper in regions of Yellowstone, employment as a truck driver in Arizona, and a motorcycle tour to Florida. He denied visiting or living in the Midwest, specifically the Ohio and Mississippi River valleys, and denied cave spelunking. He further denied visiting other potential endemic areas such as Texas, Oklahoma, and the Dakotas.

On admission, he was afebrile, saturating at 97% on room air, hypotensive with a blood pressure of 97/75, and tachycardic at 100 beats per minute, yet responsive to fluids. The physical exam was notable for a frail-appearing male with bilateral tongue nodular lesions that the patient described as painless and appeared several weeks prior to admission, see Figure 1. Due to concerns about malignancy, Oncology was consulted. Labs were significant for pancytopenia with a leukocyte count of 2.2 K/uL (NL 4–11 K/uL), hemoglobin of 12.4 g/dL (NL 13.5–17 g/dL), and platelet count of 114 K/uL (NL 130–450 K/uL). Absolute cell count showed a neutrophil count of 1.36 K/uL L (NL 1.5–7.8 K/uL) and lymphocytes of 0.49 K/uL L (NL 0.9–3.9 K/uL L). Computerized Tomography (CT) imaging of the chest, abdomen, and pelvis with contrast was significant for a left adrenal mass measuring 3.3 cm × 6 cm, see Figure 2a, which had increased from imaging 3 months prior showing a 3.2 cm × 3.5 cm mass. Imaging from 9 months ago did not show any mass. He also had a few scattered pulmonary nodules. Further work-up included Magnetic Resonance Imaging (MRI) of the abdomen and pelvis without contrast significant for diffuse thickening of the right adrenal gland up to 1 cm in the medial limb, the left adrenal mass of 6.6 cm × 3.8 cm, and splenomegaly, see Figure 2b. Due to adrenal masses, endocrinology saw the patient; he was found to have an elevated free normetanephrine level of 247 pg/mL (NL < 148 pg/mL) with a normal metanephrine range, not concerning for pheochromocytoma. The mildly elevated normetanephrine was thought to be physiologic since it was less than two times the upper limit of normal.

The tongue findings prompted a contrasted CT of the neck and soft tissue without significant findings. The patient underwent a punch biopsy of the lingual nodular lesion and a CT-guided left adrenal mass biopsy. Yeast forms compatible with *Histoplasma* were observed. On the histopathology, fungal organisms consistent with *Histoplasma* species and necrotizing granulomatous inflammation with elements compatible with *Histoplasma* were seen involving both the lingual and adrenal masses, see Figure 3, Figure 4, Figure 5 and Figure 6. These findings prompted an infectious disease (ID) consultation. Upon speaking to the family, the patient had reported episodes of confusion prior to admission, therefore, an MRI of the brain with and without contrast was obtained that showed a sub-centimeter enhancing focus within the left putamen/external capsule junction with mild edema.

Due to these findings, the patient was started on intravenous liposomal amphotericin 3 mg/kg (500 mg) every 24 h. He underwent a lumbar puncture revealing the following: cerebrospinal fluid (CSF) was clear and colorless, CSF leukocyte count of 2/µL (NL 0–5/µL), glucose 52 mg/dL (NL 40–70 mg/dL), and protein 55.7 mg/dL (NL 15–40 mg/dL). The patient was then transitioned to oral itraconazole 200 mg three times a day for 3 days followed by 200 mg twice daily. To confirm the diagnosis of histoplasmosis, tests were carried out to detect urine Histoplasma galactomannan antigen by enzyme-linked immunosorbent assay (ELISA), Histoplasma antibodies via complementary fixation (CF), and immunodiffusion (IMDF), fungal tissue culture, and tissue PCR for *Histoplasma*. The infectious disease workup is summarized in Table 1. Of note, the serological tests including Histoplasma antigens and antibodies were not available locally and sent to reference labs outside the state of Arizona. Lingual tissue was also set up for fungal cultures (set up using inhibitory mold agar, brain heart infusion agar, and mycosel agar); the calcofluor stain was positive for fungal elements, and the fungal culture yielded no growth. However, 28s ribosomal DNA PCR confirmed infection due to *Histoplasma capsulatum.*

He was discharged in stable condition on oral itraconazole 200 mg twice daily. The serum itraconazole level was <0.20 mcg/mL (Reference level: localized infection > 0.5 mcg/mL and systemic infection > 1.0 mcg/mL) and hydroxyitraconazole level was 0.36 mcg/mL after day 8 on 200 mg twice daily. Therefore, the dose was increased to 200 mg three times daily (TID), and the subsequent level improved to 0.3 mcg/mL for itraconazole and 0.8 ug/mL for the hydroxyitraconazole level. He was taken off his proton pump inhibitor since absorption of itraconazole requires an acidic pH and an EKG was obtained with a normal Qtc interval due to the risk of prolongation with itraconazole. At the one-month follow-up, the patient reported continued fatigue with some improvement, decreased tongue swelling and discomfort, and increased appetite. Subsequent itraconazole levels were therapeutic as well.

At the six-month follow-up, the patient reported significant improvement and resolution of all symptoms. Physical examination of the tongue showed resolution of the nodules and ulcerations (Figure 7). He had improvement in the repeat urine *Histoplasma* galactomannan antigen and falling *Histoplasma* yeast phase titers via CF. The *Histoplasma* antibody via immunodiffusion stayed negative. See Table 2 for a comparison of lab results. The patient had an abdominal MRI one year after his initial MRI, which showed improvement in the left adrenal mass from 3.3 × 6 cm to 2.8 × 3.7 cm (Figure 8). He is currently maintained on long-term itraconazole 200 mg TID.

The local county health department performed an environmental assessment of the patient’s residence to address the bat guano problem, to discern other related health concerns from residents, and to file a CDC Case Report Form. The assessment confirmed the presence of some bat guano around the apartment complex, as well as live bats. Although the assessment was carried out several months after onset, the same living conditions were reported by the patient during the interview. The public health case investigation also confirmed the patient’s lack of travel history outside Arizona in the prior 30 years.

## 3. Discussion

This patient’s presentation is unique due to DH in an otherwise immunocompetent patient without an obvious exposure or residence in endemic regions of histoplasmosis. Previous case reports have identified immunocompetent patients with DH; however, those patients usually have clear exposures or live in areas endemic to histoplasmosis [1]. We believe that our patient was exposed to histoplasmosis by the bat guano on his property in southern Arizona. Additionally, he may have been exposed due to his close living proximity to the cattle auction farm. Less likely exposures include his prior work as a smokejumper at Yellowstone or on his cross-country motorcycle tour. Although these potential exposures were more than thirty years ago, reactivation of the disease can occur up to fifty years after the initial infection [5].

Recently, there have been discussions centered around the accuracy of the current endemic maps for dimorphic fungi. Prevalence maps for *H. capsulatum* were created 50 years ago and changes to our environment, climate, and anthropogenic land have not been thoroughly accounted for. With the advent of immunosuppressive drugs, histoplasmosis cases have been on the rise [6]. Recent surveillance data from the CDC have shown a rise in cases not previously thought to be endemic such as in Montana, Nebraska, Minnesota, Wisconsin, and Michigan [7]. Cases with local acquisition of histoplasmosis have been found in Alberta as well [8]. As more cases are diagnosed outside of the endemic regions, the need for updating the maps will be necessary.

A retrospective analysis of >45 million Medicare fee-for-service beneficiaries from 2007 to 2016 identified an increased incidence of histoplasmosis, coccidioidomycosis, and blastomycosis outside of their historical geographical distributions. At least one county in 94% (48/51) of states was above the clinically relevant threshold (defined as 100 cases per 100,000 person-years) for histoplasmosis [9]. This further supports the idea that the endemic regions have changed since the last geographic distribution was mapped. In a study analyzing opportunities for diagnosis, the authors estimated that more than 80% of patients with histoplasmosis had at least one missed opportunity to diagnose histoplasmosis. Additionally, this study showed a 40-day delay in diagnosis with a mean of four missed opportunities to diagnose the disease. The authors found that risk factors for a delay in diagnosis are prior antibiotic use and history of prior pulmonary disease [10]. Diagnostic delay in disseminated mycosis had a major clinical impact in 66% of cases [9]. Limiting our diagnosis based on endemic regions could pose a threat to patient outcomes, such as our patient who presented with DH. He had known adrenal masses for 3 months that prompted a malignancy work-up before we were able to consider histoplasmosis based on the tongue lesions.

Risk factors for DH include patients with HIV disease, immunosuppressive therapies including steroids and TNF-blockers, hematologic malignancies, and transplant recipients [11]. Symptoms include fever, fatigue, weight loss, shortness of breath, and diarrhea and the physical exam will show lymphadenopathy, hepatomegaly or splenomegaly, skin manifestations, and oral lesions. Laboratory findings are significant for pancytopenia, transaminitis, and high lactate dehydrogenase. In addition to an unclear epidemiologic exposure to histoplasmosis, our patient did not have any traditional risk factors that would classify him as an immunocompromised host. DH commonly affects the lungs, liver, spleen, GI tract, bone marrow, and central nervous system (CNS) with infiltration of the adrenal glands less commonly seen [12]. DH involves the CNS in 5–10% of patients with about one-third of those patients being immunocompetent. CNS histoplasmosis causes abnormal brain imaging in 72% of those patients and commonly leads to headaches, confusion, neck stiffness, altered mental status, and focal deficits. CSF may show a lymphocytic predominance but may also be bland [13].

The initial concern for our patient was malignancy in the setting of adrenal masses and pancytopenia. Oropharyngeal and laryngeal lesions of the tongue, hard and soft palate, and buccal mucosa are commonly seen in DH; however, patients are usually immunocompromised [14]. We performed a fourth-generation HIV test to consider this as a susceptibility factor; however, the test was negative. Additionally, the patient was tested for coccidioidomycosis given Arizona is an endemic area for coccidioidomycosis, but this was also negative. Often, these lesions are confused with head and neck malignancy, especially in patients without risk factors in non-endemic regions of histoplasmosis [15]. Pincelli et al. (2019) published a case series of ten patients with DH presenting with oral histoplasmosis. They identified four patients with DH who had adrenal involvement, three of which had a smoking history but were not immunocompromised. The fourth patient was immunocompromised on rituximab, fludarabine, and bendamustine therapies for lymphocytic leukemia/lymphoma but did not have a smoking history. This series also noted a male predominance in their patient population. All their patients had lived in an endemic region of the US at some point [14]. Isolated mucocutaneous presentation of histoplasmosis in non-HIV patients is rare and is often confused with malignancy. As for pancytopenia, there is one case report of DH in an immunocompetent patient with bilateral adrenal masses that presented with pancytopenia. Bone marrow biopsy confirmed histoplasmosis involvement, and the patient had a return of normal blood count after treatment [16]. We would have conducted a bone marrow biopsy on our patient if his pancytopenia did not improve.

Adrenal involvement is frequently found in the biopsy but only causes adrenal necrosis in less than 10% of DH patients [17]. Taking into consideration what we now know about histoplasmosis, the evaluation of this patient should have included histoplasmosis in the setting of adrenal involvement with oral lesions. Since he had not recently resided in an endemic region and was not immunocompromised, this diagnosis was delayed. In the future, it would be important to keep a broad differential even if patients do not meet the traditional epidemiology as the current literature may not be up-to-date.

There are several ways of diagnosing histoplasmosis including tissue pathology, fungal cultures, antigen testing, antibody testing, and molecular testing [12]. Histopathology, direct microscopic identification, or cultures confirm the diagnosis of *H. capsulatum.* The stains to identify the fungi include Wright, periodic acid-Shiff, or Grocott–Gomori’s methenamine silver of tissue biopsies, skin scrapes, or bone marrow smears [18]. In the immunocompromised, DH can be fatal and is treated with liposomal amphotericin B and itraconazole [19]. The Infectious Diseases Society of America (IDSA) guidelines for DH recommend starting liposomal amphotericin B (3 mg/kg daily) in patients with severe illness. Liposomal amphotericin B is then transitioned to 200 mg itraconazole twice daily for 1 year once the patient is stable. Itraconazole can be lifelong if there is a DH relapse [20].

Itraconazole blood levels are measured after 7–10 days of therapy initiation to determine therapeutic levels and avoid toxicity. The minimum inhibitory concentration for 90% of *H. capsulatum* strains is 1.06 mcg/mL and the therapeutic concentration is >1–2 mcg/mL as measured by the bioassay or the sum of the parent itraconazole drug and the hydroxyl metabolite measured by high-performance liquid chromatography (HPLC). Response to treatment is monitored via serum and urine *Histoplasma* galactomannan antigen levels [20].

## 4. Conclusions

We presented an immunocompetent patient from southern Arizona with DH involving the lungs, adrenal glands, tongue, and CNS. We believe that our patient was exposed to histoplasmosis by bat guano on his property in southern Arizona. According to the state and county health department data, this is likely one of the first autochthonous cases of histoplasmosis in Arizona. Interestingly, *H. capsulatum* was isolated from bats in southern Arizona in 1967 [21]. His diagnosis was delayed due to the differential diagnosis not including histoplasmosis. Further research involving the endemic regions of this dimorphic fungus revealed maps that have not been updated in 50 years placing into question the accuracy of the word “endemic” as it relates to regions where particular fungi can be found. Physicians should be aware of the possibility of histoplasmosis in Arizona and consider the diagnosis in patients with clinically compatible illnesses.

## Figures and Tables

**Figure 1 jof-10-00756-f001:**
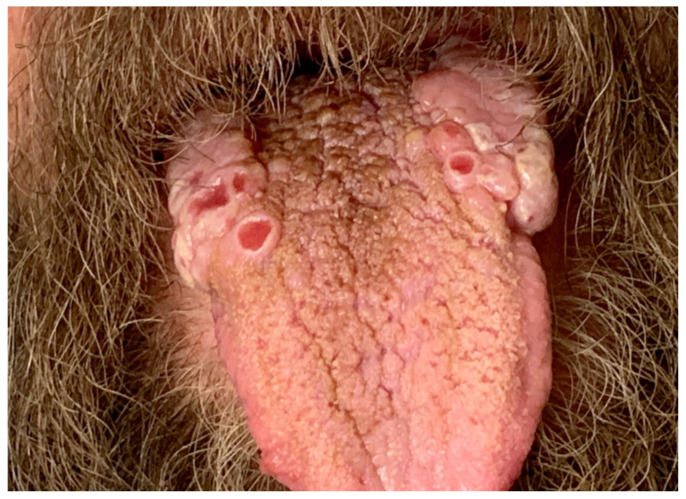
Bilateral tongue nodules and ulcerations found on the physical exam.

**Figure 2 jof-10-00756-f002:**
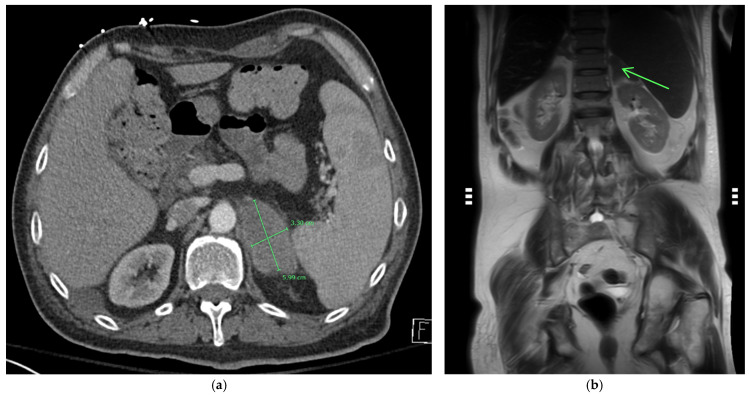
(**a**) CT abdomen with contrast showing a large 3.3 × 6 cm enhancing left adrenal mass. (**b**) MRI imaging without contrast of the left adrenal mass demonstrates low T1 and T2 signals on the coronal MRI images (see arrow).

**Figure 3 jof-10-00756-f003:**
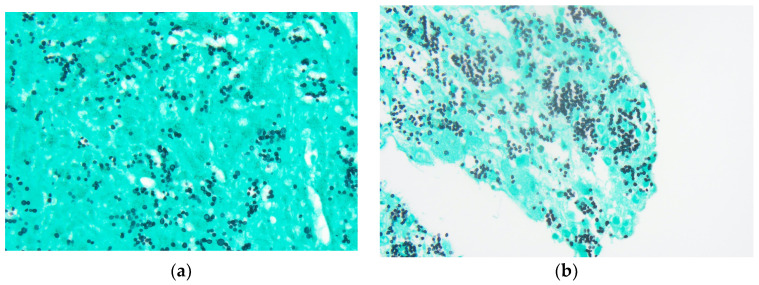
Adrenal mass biopsy: (**a**) Grocott methenamine silver (GMS) stain, 600× magnification. Organisms show little variability in size and display narrow-based budding; (**b**) GMS stain, 600× magnification.

**Figure 4 jof-10-00756-f004:**
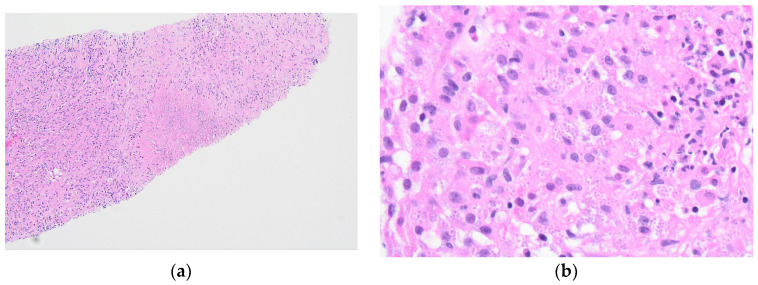
Adrenal mass biopsy: (**a**) Granulomatous inflammation—Necrotizing granulomatous inflammation is present (hematoxylin and eosin stain (H&E), 100× magnification); (**b**) Intracellular organisms—Numerous monomorphic organisms are seen within histiocytes (hematoxylin and eosin stain, 600× magnification).

**Figure 5 jof-10-00756-f005:**
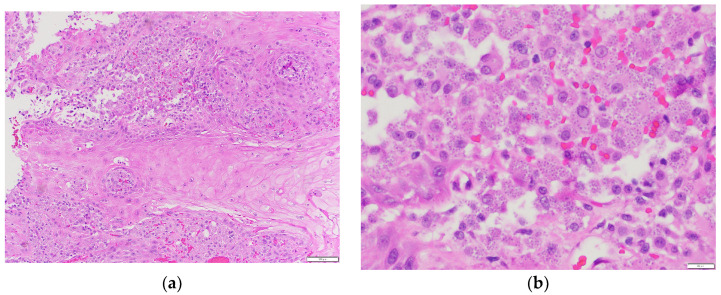
Tongue punch biopsy: (**a**) Histoplasmosis 100× H&E. Low power view of abundant histiocytes with numerous intracytoplasmic Histoplasma organisms; (**b**) Histoplasmosis 400× H&E. High power view of histiocytes with Histoplasma organisms.

**Figure 6 jof-10-00756-f006:**
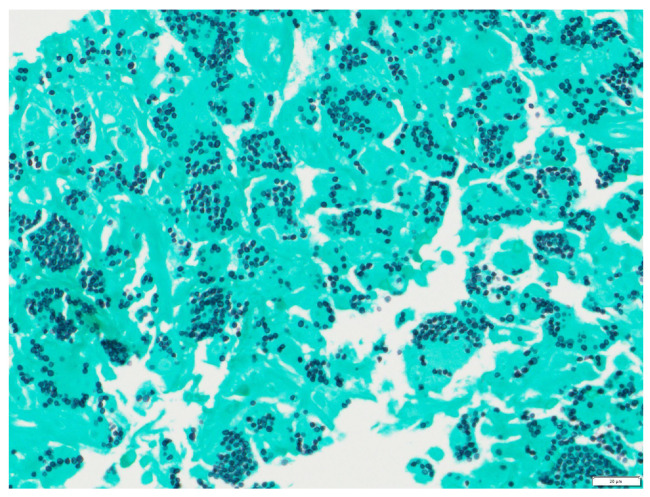
Tongue punch biopsy: Histoplasmosis 400× GMS. Grocott methenamine silver (GMS) stain highlighting Histoplasma cell walls within histiocytes.

**Figure 7 jof-10-00756-f007:**
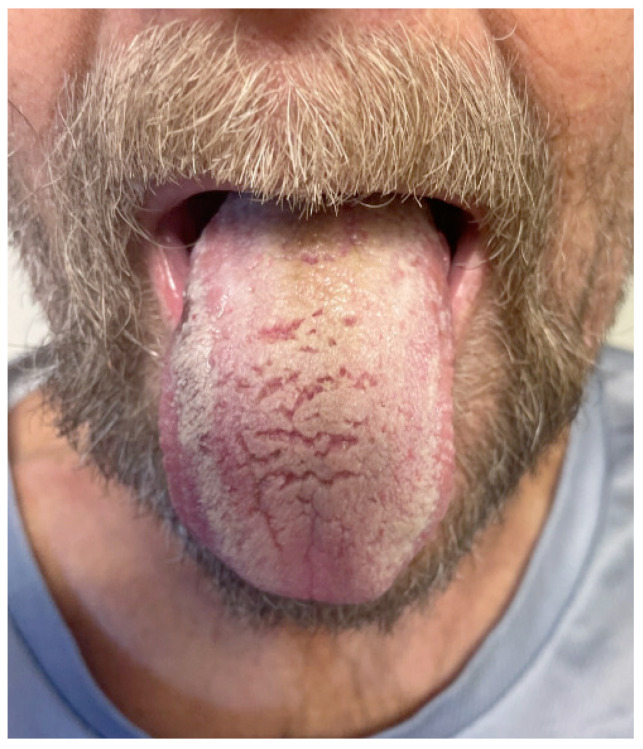
Improvement of bilateral tongue nodules and ulcerations found on the 6-month follow-up physical exam.

**Figure 8 jof-10-00756-f008:**
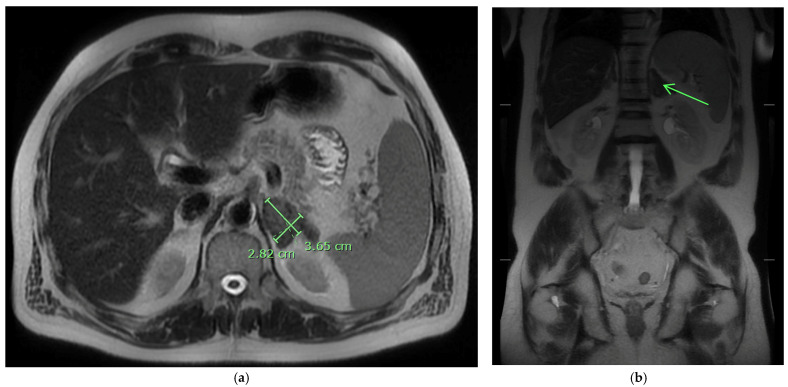
(**a**) MRI abdomen pelvis w/out contrast showing 2.8 × 3.7 cm enhancing left adrenal mass at 1-year follow-up. (**b**) MRI imaging without contrast of adrenal mass demonstrates low T1 and T2 signals on the coronal MRI images at 1-year follow-up (see arrow).

**Table 1 jof-10-00756-t001:** Results of ID initial work up.

Diagnostic Tests	Results
Urine *Histoplasma* galactomannan antigen	3.27 ng/mL (NL < 0.20 ng/mL)
Serum *Histoplasma* galactomannan antigen	1.14 ng/mL (Positive: 0.20–20.00 ng/mL)
*Histoplasma* yeast phase antibody via compliment fixation	1:32 (reference interval < 1:8: not detected)
*Histoplasma* mycelia phase antibody via compliment fixation	<1:8 Interpretive Criteria: <1:8 Antibody Not Detected > or = 1:8 Antibody Detected
*Histoplasma* antibody via immunodiffusion	Antibody not detected
28S ribosomal DNA PCR (on tongue tissue)	Positive for *Histoplasma capsulatum*
Fungal culture (tongue)	No mold or yeast isolated, fungal elements resembling yeast on fungal stain
Coccidioides antibody via EIA (IgG)	Indeterminate
Coccidioides complement fixation titer (IgG)	Negative
Fourth-generation HIV-1/2 antibody and HIV-1 antigen combination assay	Negative
Rapid plasma reagin test	Negative
Beta-D-glucan	269 pg/mL (normal reference < 80)

**Table 2 jof-10-00756-t002:** Comparison of admission labs versus labs at the 6-month appointment.

Diagnostic Tests	On Admission	6-Month Follow-Up
Urine Histoplasma Galactomannan antigen	3.27 ng/mL (NL < 0.20 ng/mL)	<0.2 ng/mL
Histoplasma yeast phase antibody via CF	1:32 (Normal values not provided)	1:16
Histoplasma mycelial phase antibody via CF	<1:8 Interpretive Criteria: <1:8 Antibody Not Detected > or = 1:8 Antibody Detected	<1:8
Histoplasma AB via IMDF	Negative	Negative
Serum Histoplasma galactomannan antigen	1.14 ng/mL	Not conducted

## Data Availability

The original contributions presented in the study are included in the article, further inquiries can be directed to the corresponding author.

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
