# Peer review of "Disseminated Histoplasmosis in an Immunocompetent Patient from Southern Arizona"

_jof, 2024, doi:10.3390/jof10110756_

Round 1
Reviewer 1 Report (Previous Reviewer 2)
Comments and Suggestions for Authors
Dear Authors,
Thank you for making the requested changes to the manuscript.
Please be consistent with the name of histoplasma test used since there are many and can be confusing for the reader, e.g. "He had improvement in the repeat urine Histoplasma Galactomannan antigen. The Histoplasma antibody, immunodiffusion, and the Histoplasma capsulatum H antibody were negative. There was also improvement in the Histoplasm Yeast titer ..."
-for urine antigen use always "urine Histoplasma Galactomannan antigen"
-specify which Histoplasma antibody you refer to
-immunodiffusion appears only once in the text (discussion) and it is unclear what you refer to, probably Histoplasma H and M antibody, give always a full name, with type of assay (immunodiffusion) used
-provide in table 2 all repeated tests, even if negative
It could be interesting to know if all the microbiological tests were available locally or were sent externally
Author Response
Comments 1: “Please be consistent with the name of histoplasma test used since there are many and can be confusing for the reader, e.g. "He had improvement in the repeat urine Histoplasma Galactomannan antigen. The Histoplasma antibody, immunodiffusion, and the Histoplasma capsulatum H antibody were negative. There was also improvement in the Histoplasm Yeast titer ..."
“For urine antigen use always "urine Histoplasma Galactomannan antigen” ”
Response 1: This has been properly changed throughout the paper. See lines 107-118, 134-139, and line 254.
Comments 2: “Specify which Histoplasma antibody you refer to”
Response 2: These include Histoplasma Ab via complement fixation and immunodiffusion. The text has been included to clarify this point. See Table 2
Comments 3: “Immunodiffusion appears only once in the text (discussion) and it is unclear what you refer to, probably Histoplasma H and M antibody, give always a full name, with type of assay (immunodiffusion) used. provide in table 2 all repeated tests, even if negative”
Response 3: Thanks for pointing this out. We have included the IMDF assay in table 2 and it was negative.
Comments 4: “It could be interesting to know if all the microbiological tests were available locally or were sent externally”
Response 4: None of the microbiological tests were available locally, they were sent to reference labs in Indiana for the serological testing and to Washington state for PCR testing. See lines 112-113
Reviewer 2 Report (Previous Reviewer 3)
Comments and Suggestions for Authors
This is a resubmission of a single case report of disseminated histoplasmosis in a patient from Southern Arizona. The case reveals a classic case of subacute disseminated histoplasmosis. The distinguishing feature is the georgraphic setting otherwise, there is nothing that distinguishes this report from many others.
Author Response
Comment 1” “This is a resubmission of a single case report of disseminated histoplasmosis in a patient from Southern Arizona. The case reveals a classic case of subacute disseminated histoplasmosis. The distinguishing feature is the geographic setting otherwise, there is nothing that distinguishes this report from many others.”
Response 1: “There are a variety of factors that make this case unique. The first is that this immunocompetent patient likely contracted histoplasma from bat guano in Arizona. While many papers address immunocompetent patients with Disseminated Histoplasmosis, most of these have obvious exposures to endemic areas. Our patient had not traveled to endemic areas within 30 years making this a unique case presentation. The patient was initially worked up for malignancy given his adrenal mass and significant weight loss. It was not until adrenal biopsy showed histoplasma that Infectious disease was consulted. Being in Arizona with no recent travel, his diagnosis was delayed. It is critical that case reports like these are published to broaden the infectious disease differential for physicians practicing outside of endemic areas to prevent further delay in the diagnosis of similar patients.
Reviewer 3 Report (New Reviewer)
Comments and Suggestions for Authors
The case report by Monique Coawford and et al. present a disseminated histoplasmosis in the inmunocompetent host outside of the know endemic regions, with una “atypical” clinical presentation. It is interesting to publicize this type of infection to open the diagnosis and take into account other options that would be ruled out in relation to infectious diseases. A rapid diagnosis is crucial to prescribe the appropriate treatment in the patient.
Minor comments:
· Complete Figure 6
· Table 1 and 2 provide information about the serological tests used, PCR assay, culture medium, etc.
Are field studies done to show that these areas would be infected?
·
·
Comments on the Quality of English LanguageI suggest to the authors a review of English
Author Response
Comment 1: “Complete Figure 6”
Response 1: At this time, we only have 400X photomicrograph available.
Comment 2: “Table 1 and 2 provide information about the serological tests used, PCR assay, culture medium, etc.”
Response 2: Serological tests have been described and comments included re: culture medium discussed as well. See table 1 and 2.
Comment 3: “Are field studies done to show that these areas would be infected?”
Response 3: The primary reasons that the field studies were done were to address the bat guano problem, to discern other related health concerns from residents, and to file a CDC Case Report Form. This has been included in the last paragraph of the Case Presentation. Cochise County completed the case interview, as well as an environmental assessment of the residence. The Arizona department of health for vector-borne & Zoonotic Diseases (VBZD) & Mycotic Diseases is still looking for options for possible bat/environmental testing.
Comment 4: “I suggest to the authors a review of English”
Response 4: We have thoroughly edited this paper and made some grammatical changes.
Reviewer 4 Report (New Reviewer)
Comments and Suggestions for Authors
The manuscript “Disseminated Histoplasmosis in an Immunocompetent Patient from Southern Arizona” presents the case of an immunocompetent patient from Southern Arizona with disseminated histoplasmosis, probably acquired from bat guano in Southern Arizona. It is an interesting article that shows the importance of finding histoplasmosis in non-endemic areas, I only have a few comments.
Line 101. Mention that yeast-like forms compatible with Histoplasma were observed.
Line 129. Mention that, to confirm the diagnosis of histoplasmosis, tests were carried out to detect antigen (ELISA), antibodies (Complementary fixation reaction, gel immunodiffusion), culture and, PCR for Hstoplasma, Coccidioides.
In Table 2, what technique was used to detect HIV?
In the discussion, the authors do not mention why they performed tests to detect Coccidioidomycosis and HIV, in the first case because Arizona is an endemic area for coccidioidomycosis and in the second case to verify if the patient had HIV and consider it as a susceptibility factor?
Line 331. Add to bibliography: Turkish Journal of Pathology. 39(2), 133-139
Author Response
Comment 1: “Line 101. Mention that yeast-like forms compatible with Histoplasma were observed.”
Response 1: We have included this in line 93
Comment 1: “Line 129. Mention that, to confirm the diagnosis of histoplasmosis, tests were carried out to detect antigen (ELISA), antibodies (Complementary fixation reaction, gel immunodiffusion), culture and, PCR for Hstoplasma, Coccidioides.”
Response 1: We have included this sentence of clarification in line 107.
Comment 2: “In Table 2, what technique was used to detect HIV?”
Response 2 The HIV screen is reported in Table 1. We have clarified that this was a 4th generation HIV-1/2 antibody and HIV-1 antigen combination assay.
Comment 3: “In the discussion, the authors do not mention why they performed tests to detect Coccidioidomycosis and HIV, in the first case because Arizona is an endemic area for coccidioidomycosis and in the second case to verify if the patient had HIV and consider it as a susceptibility factor?”
Response 3: We updated the discussion to include the reasoning for testing HIV and coccidioidomycosis. See lines 210-213.
Comment 4: “Line 331. Add to bibliography: Turkish Journal of Pathology. 39(2), 133-139”
Response 4: We updated the bibliography on line 350 to include the volume and page number.
Round 2
Reviewer 2 Report (Previous Reviewer 3)
Comments and Suggestions for Authors
Despite what the authors argue, finding a case of disseminated histoplasmosis in a so-called immunocompetent individual in Arizona is not sufficient grounds to be considered unique. Second, the authors never explore whether this individual has a genetic mutation that might render him susceptible. Aside from this, a single case report on a patient from Arizona lacks general interest.
This manuscript is a resubmission of an earlier submission. The following is a list of the peer review reports and author responses from that submission.
Round 1
Reviewer 1 Report
Comments and Suggestions for Authors At least 6 months follow-up is needed. Would add that to the manuscript. Exclude symptoms on changing vocabulary from endemic to dimorphic as it may never occur. Included references from Marciano BE and Holland SM on immunodeficiencies predisposing to histo and other endemic mycoses.Provide data supporting 25-45% of cases have oral involvement. Unlikely that adrenal glands are the most common organ involvement outside the lungs, exclude or provide reference. Delete sentences regarding endemic pattern of histoplasmosis without supporting data. Provide data on renal involvement. Provide data on CNS involvement. Provide IDSA treatment guideline recommendations. Delete comments that "consequences of the current literature can lead to delays in diagnosis and treatment…" as there is no supporting evidence
Reviewer 2 Report
Comments and Suggestions for Authors
This is an interesting report of a single case of disseminated histoplasmosis in an immunocompromised patient outside the endemic region.
My major comments are:
In order to be more applicable to the international population, the authors could provide a map showing the currently considered endemic region and the places where the patient travelled.
Moreover, it would be useful to mention in the discussion if there are other cases published of histoplasmosis outside the endemic regions worldwide, in order to substantiate the authors’ claim that the maps are outdated.
The statement “Recent research has suggested that more cases are being seen outside of endemic regions increasing the concern for a missed or delayed diagnosis” should be supported by also papers other than ref 5.
Ref 5 should be discussed more in detail.
The specific assays used for diagnostic tests reported in table 2 should be provided for histoplasma test, and for quantitative test, the normal values should be provided.
Minor
Not all results are reported in SI
The dose of liposomal amphoteric B is usually reported per Kg – please provide the data.
There are several repetitions, including the visited regions, which could be provided more generically (the major point)
Discussion could be shortened and more focused.
Reviewer 3 Report
Comments and Suggestions for Authors
The manuscript by Crawford and colleagues reports a patient whose primary residence is in Arizona and who developed disseminated histoplasmosis. The diagnosis was delayed largely because, it seems, of unfamiliarity with this fungal infection in Arizona. The case is interesting but far from novel. The patient had been to potential sites of exposure, particularly Florida. Yellowstone is certainly a possibility but less likely than Florida. There is no information whether the patient had been to other potential areas such as east Texas, Oklahoma, or the Dakotas.
The purpose of this case report is unclear. The literature is replete with cases of disseminated histoplasmosis in many different individuals and in many different settings. Not every case needs to be reported in the literature. There is nothing in this case description that distinguishes it from the many others populating the literature. The authors have not indicated the novelty. The fact that the patient resided in Arizona is not a sufficiently novel finding any more than if the patient resided in Washington State, Oregon or Utah. Perhaps if the patient had been found to have an underlying immune defect that might elevate slightly the importance of this manuscript. But as it written, there is little that differentiates it from many other case reports.